# STABILITY ANALYSIS OF VARIOUS SYMBOLIC RULE EXTRACTION METHODS FROM RECURRENT NEURAL NETWORK

## ABSTRACT

This paper analyzes two competing rule extraction methodologies: quantization and equivalence query. We trained 3600 RNN models, extracting 18000 DFA (Deterministic Finite Automata) with a quantization approach (k-means and SOM) and 3600 DFA by equivalence query($L^*$) methods across 10 initialization seeds. We sampled the datasets from 7 Tomita and 4 Dyck grammars and trained them on 4 RNN cells: LSTM, GRU, O2RNN, and MIRNN. The observations from our experiments establish the superior performance of O2RNN and quantization-based rule extraction over others. $L^*$, primarily proposed for regular grammars, performs similarly to quantization methods for Tomita languages when neural networks are trained completely. However, for partially trained RNNs, $L^*$ shows instability in the number of states in DFA, e.g., for Tomita 5 and Tomita 6 languages, $L^*$ produced more than 100 states. In contrast, quantization methods result in rules with the number of states very close to ground truth DFA. Among RNN cells, O2RNN produces stable DFA consistently compared to other cells. For Dyck Languages, we observe that although GRU outperforms other RNNs in network performance, the DFA extracted by O2RNN has higher performance and better stability. The stability is computed as the standard deviation of accuracy on test sets on networks trained across 10 seeds. On Dyck Languages, quantization methods outperformed $L^*$ with better stability in accuracy and the number of states. $L^*$ often showed instability in accuracy in the order of $16\% - 22\%$ for GRU and MIRNN while deviation for quantization methods varied in $5\% - 15\%$. In many instances with LSTM and GRU, DFA's extracted by $L^*$ even failed to beat chance accuracy ($50\%$), while those extracted by quantization method had standard deviation in the $7\% - 17\%$ range. For O2RNN, both rule extraction methods had a deviation in the $0.5\% - 3\%$ range.

## 1 INTRODUCTION

Stateful networks such as recurrent networks (RNNs) effectively model complex temporal and noisy patterns on real-world sequences. In particular, RNNs with minimal syntactic grammatical errors can effectively recognize grammatical structure in sequences. This is evident by their ability to generate and generalize structure data, such as language, source code (C, C++, Latex, etc.), and program synthesis. To better understand the internal workings of RNNs, recent works are focused on extracting interpretable rules from trained RNNs that better evaluate the ability of RNNs to recognize grammatical structures. However, the stability of these rule extraction needs to be better studied, as instability often leads to failure in extraction. This work aims to explain the internal workings of various types of RNNs. We mainly analyze their stability through rigorous comparison to fundamental concepts in formal languages, namely finite automata and regular languages.

Recent theoretical works have shown that with some constraints, such as finite precision and time, RNNs and GRU can only recognize regular grammars Merrill (2019). However, another class of RNNs, 02RNN, has a much more robust construction and has shown equivalence with only $N + 1$ state neurons for DFA with $N$ states and is argued to be stable Mali et al. (2023a). This paper thoroughly analyzes the stability of the internal state of various types of RNNs, such as first-order (RNN, GRU, LSTM) and second-order RNN (O2RNN). One family of approaches widely used in

literature to understand the internal workings of RNNs is to extract an interpretable state machine from the trained RNN. Two widely used approaches that take inspiration from grammatical induction are L-star ANGLUIN (1987); Weiss et al. (2018) and Quantization-based approaches Giles & Omlin (1993); Omlin & Giles (1996b). L-star is an active learning approach that learns via two forms of queries. The first is the membership query, which focuses on strings in language L, and the other is the equivalence query, which compares a candidate DFA to the true or oracle DFA. However, L-star requires RNNs to be 100% accurate, which is problematic in a real-world scenario as NNs rarely get achieve 100%. Second, they also assume these black box models that compute oracles to be available at train time, which is again problematic for large, expensive equivalence queries. Other works are based on the quantization approach, which uses clustering (k-means, SOM) on the RNN hidden states to extract the state machines and does not suffer from the above problem. However, none of the prior work analyzes the stability of these rule extraction methods, as NNs are stochastic and introduces various forms of uncertainty resulting in generalization issue. Thus, this also hampers the extraction strength of various extraction approaches, resulting in sub-optimal state machines.

**This Work** We conduct an extensive empirical study with the aim of discovering a better extraction approach that is stable across a wide variety of RNNs. This work also sheds light on whether the theoretical limitations of RNNs hold in practice when trained using backpropagation of error through time. We make the following main contributions:

1. We show that quantization-based rule extraction is more stable than the equivalence query-based method.

2. Quantization-based approach performs better even when RNNs cannot reach $100\%$ accuracy.

3. Quantization-based approach performs better when underlying grammar contains complex patterns.

4. We empirically show that $2^{nd}$ RNNs encapsulate stable rules than $1^{st}$ order networks. Thus empirically validating prior theoretical work.

5. We conduct extensive experiments across different RNN architectures with varied hidden state sizes and grammar types to show the stability of the rule extraction method.

## 2 RELATED WORK

RNNs are known to be excellent at recognizing patterns in text and algorithmic patterns. Extensive work is focused on deriving computational power and expressivity of these systems. For instance, memory-less RNNs with infinite precision are known to be Turing complete Siegelmann & Sontag (1995); Mali et al. (2023a), whereas memory-augmented RNNs are Turing Complete even with finite precision Stogin et al. (2020); Mali et al. (2023b). Funahashi & Nakamura (1993) showed there exists a class of RNN that can approximately represent dynamic systems, whereas Rabusseau et al. (2019) showed that linear second-order RNN is equivalent to weighted automaton.

Recently Merrill et al. (2020) classified NNs based on space complexity and rational recurrence. They show that GRU and RNN can model regular grammars with $O(1)$ while LSTM can also model counter machines with space complexity of $O(\log n)$ where $n$ is the input sequence length.Hewitt et al. (2020) show that the theoretical bound of $O(m \log k)$ for a Dyck-$(k, m)$ language on the size of hidden units is too tight and is not achievable even by unbounded computation. Dyck-$(k, m)$ languages have $k$ types of nested parenthesis with a maximum depth of $m$ nests. The best empirical results are obtained at $O(k^{\frac{m}{2}})$.

Recent works are more focused on exploring the relationship between RNN internals and DFAs through a variety of methods. One such direction is based on automata extraction and grammatical induction. ANGLUIN (1987) proposed an equivalence query-based approach, $L^*$, to construct DFA (Deterministic Finite Automata) from a set of given examples. The method requires an oracle to classify positive and negative examples also provide counterexamples. Weiss et al. (2018) used RNN models as oracles to $L^*$ algorithm. Giles & Omlin (1993) and Omlin & Giles (1996b) showed quantization-based extraction mechanisms for symbolic rules from Recurrent Neural Networks in the form of DFAs. This was further supported by Wang et al. (2017) with empirical results. All these prior results primarily focus on Tomita Grammars/LanguagesTomita (1982). Tomita languages have since been used to demonstrate the computational capacity of neural networks for regular grammars.

Sennhauser & Berwick (2018); Suzgun et al. (2019a); Mali et al. (2021a) provided the empirical evaluation of LSTM on Context-Free Grammars. Suzgun et al. (2019a); Mali et al. (2021a) show that LSTM with just one hidden unit is sufficient to learn Dyck-1 grammar. Bhattamishra et al. (2020b) experimented with self-attention models to demonstrate their ability to learn counter languages like $n$-ary Boolean Expressions. Ebrahimi et al. (2020) showed that networks with self-attention perform better than LSTMs in recognizing Dyck-$n$ languages. Suzgun et al. (2019b); Mali et al. (2021b) trained memory augmented RNNS to learn Dyck Languages with up to 6 parenthesis pairs. Mali et al. (2020) improved upon the memory-augmented Tensor RNNs and used an external differentiable stack to recognize long strings of Context-Free Grammars. Recently Delétang et al. (2022) conducted the most comprehensive study, which places several types of NNs based on the Chomsky hierarchy. However, their work was primarily focused on first-order NNs.

Prior works have largely focused on the idea of learnability of regular and context-free grammars by Recurrent Neural Networks and some other neural networks. Theoretical bounds, empirical bounds, and rule extraction methods remain the main focus of research. The question of the empirical stability of these models and the rules extracted from them have largely gone unanswered. In this paper, we perform large-scale experiments and report our findings on the stability of RNNs and the two rule extraction methods (quantization and equivalence query) on Tomita and Dyck languages.

## 3 BACKGROUND & METHODOLOGY

**Recurrent Neural Networks** To evaluate the stability of extraction methods, we train four types of RNN models. We use LSTM (Hochreiter & Schmidhuber, 1997) and GRU (Chung et al., 2014) for $1^{st}$ order networks and O2RNN (Giles et al., 1989) and MIRNN (Wu et al., 2016) as $2^{nd}$ order networks. We briefly describe the state update equation in O2RNN which has a higher order operation facilitated by a $2^{nd}$ order weight tensor given as follows:

$$h_i^{(t+1)} = \sigma(\sum_{j,k} W_{ijk} x_i^{(t)} h_k^{(t)} + b_i)$$

, where $h$ is the state transition of the O2RNN such that $h_t \in \mathbb{Q}$ and $\mathbb{Q}$ is a set of rational numbers, $W \in \mathbb{R}^{n \times n \times n}$ contains the $2^{nd}$ order recurrent weight matrices, $b \in \mathbb{R}^n$ are the biases, $sigma$ is the nonlinear activation function such as the logistic sigmoid, and $x$ is the input symbol at time $t$. It is theoretically shown that 02RNN can simulate DFA with finite precision and time with only $n + 1$ state neurons.

**Theorem 3.1.** *Mali et al. (2023a) Given a DFA $M$ with $n$ states and $m$ input symbols, there exists a $k$-neuron bounded precision O2RNN with sigmoid activation function ($h_H$), where $k = n + 1$, initialized from an arbitrary distribution, that can simulate any DFA in real-time $O(T)$.*

Furthermore, it was proved that the above construction using O2RNN is stable. This is important in our study as we empirically try to show the stability when models are trained using gradient-based learning. On the other hand, MIRNN approximates the second-order connection by introducing rank-1 approximation in the form of multiplicative integration (Wu et al., 2016). This is achieved by replacing multiplication with Hadamard product in the vanilla RNN equation. Thus the state transition or update equation for MIRNN is written as follows:

$$h^{(t+1)} = \tanh(\alpha U x^{(t)} \odot V h^{(t)} + \beta_1 U \odot x^{(t)} + \beta_2 V \odot h^{(t)} + b)$$

where $\alpha$, $\beta_1$ and $\beta_2$ are trainable parameters or can be considered as additional sets of bias vectors, and $\odot$ denotes Hadamard product.

**Deterministic finite automaton** A Deterministic Finite Automaton is a 5-tuple $(Q, \Sigma, \delta, q_0, F)$. $\Sigma$ is the set of symbols called alphabet which are used to create strings. A DFA can either accept or reject an input string. A language recognized by DFA is a set of strings accepted by the DFA. Grammar is a set of rules followed by the language. DFA acts as a machine that checks if a string follows the grammatical rules. Hence, DFA are considered to be the representation of grammatical rules.

$Q$ is the set of all states in the DFA. $q_0 \in Q$ is the initial state of the DFA and $F \subseteq Q$ is the set of final accepting states. Any state $q \in Q$ s.t. $q \notin F$ is a rejecting state. The transition matrix $\delta : Q \times \Sigma \to Q$ encodes the rules of state transitions.

**DFA Extraction**   There are two prominent techniques for DFA extraction from RNNs: quantization-based (Omlin & Giles, 1996b)(Giles & Omlin, 1993) and counter-example based (ANGLUIN, 1987)(Weiss et al., 2018).

In a quantization-based approach, hidden states $(h_{ij})$ from each step $(j)$ of RNN computation for sample $(i)$ are collected and clustered into $k$ groups $G = \{g_1, g_2, ...g_k\}$. Here we can choose a clustering algorithm $C$ such that $C(h_{ij}) = g_m \in G$. These groups form one-to-one correspondence with the states of the DFA $(Q = G)$. A transition matrix $(\delta(C(h_{ij}), a_{ij}) \to g_n \in G)$ is then created by mapping group transitions for each input $a_{ij} \in \Sigma$. Initial state $q_0 = C(h_{init})$ and final states $(F)$ are selected through majority voting of qualified inputs. This gives us an initial DFA, which is then minimized (Hopcroft, 1971) to get a final minimal DFA. In this work, we use k-means (Lloyd, 1982) and self-organizing maps (SOM) (Kohonen, 1982) for clustering. For each model we extract 5 DFA's with a successively increasing number of states. These DFA's are minimized, and the best performing DFA (on validation set) is selected. With 3200 models and 5 extraction per model, we extract 16000 DFAs.

$L^*$ proposed by (ANGLUIN, 1987) creates a transition function of states by repeatedly asking an oracle to classify strings. The oracle is also responsible for providing counter-examples. It is discussed in detail in Appendix B. (Weiss et al., 2018) use RNN models as oracle for $L^*$. We follow their approach to extract DFA using $L^*$ for each of the 3200 models. One observation we made here is that the extraction heavily depends on initial counter-examples. This does not prove good for Dyck grammars especially.

**Language from a Concentric Ring Representation**   The concentric ring representation is used to analyze the complexity of a language Watrous & Kuhn (1991). It reflects the distribution of its associated positive and negative strings within a certain length. In each concentric ring, we can observe that all strings with the same length are arranged in a lexicographic order where white represents positive string and black/blue represents negative strings. The more variation one observes in a concentric ring, the more difficulty neural networks have in learning. In figure 1 and 2 we show the ring presentation for Tomita and dyck grammar, respectively.

**Stability Analysis**

**Theorem 3.2.** *BROUWER'S FIXED POINT THEOREM (Boothby, 1971): For any continuous mapping $f : Z \to Z$, where $Z$ is a compact, non-empty convex set, $\exists z_f$ s.t. $f(z_f) \to z_f$*

**Definition 3.3.** *Let $\mathcal{E}$ be an estimator of fixed point $z_f$ for mapping $f : Z \to Z$, where $Z$ is a metric space. $\mathcal{E}$ is a stable estimator iff estimated fixed point $\hat{z}_f$ is in immediate vicinity of $z_f$, i.e. $|\hat{z}_f - z_f| < \epsilon$, for arbitrarily small $\epsilon$.*

**Corollary 3.4.** *Stability of estimator $\mathcal{E}$ can be shown by iteratively computing $z^{(t+1)} = \mathcal{E}_f(z^{(t)})$ with $z^{(0)}$ in the neighborhood of $z_f$. For stable estimator $\mathcal{E}$, $\lim_{t \to \infty} z^{(t)} = z_f$. Neighborhood of $z_f$, $N_{z_f}$ is a set of points $z_n \in N_{z_f}$ near $z_f$ s.t. $|z_n - z_f| < \delta$.*

Omlin & Giles (1996a) use this idea of stability to show that for provably stable second-order RNNs, the sigmoidal discriminant function should drive neuron values to near saturation. Similarly, Stogin et al. (2020) use the fixed point analysis to prove the stability of differentiable stack and differentiable tape with RNNs.

**Definition 3.5.** *Let $\mathcal{E}^*$ be a stochastic fixed point estimator for $f$. The estimated $RMS$ error for fixed point estimation, calculated iteratively as $\hat{z}_f^{(i)} = \mathcal{E}_f^*(\hat{z}_f^{(i-1)})$ can be defined as:*

$$e_{rms} = \sqrt{\frac{1}{n} \sum_{i=0}^{n} ||\hat{z}_f^{(i)} - z_f||^2}$$

*Note that when true $z_f$ is unknown, the law of large numbers allows us to use the estimated mean $\overline{z}_f$ instead. Also the error estimate $e_{rms}$ is the standard deviation from $z_f$*

**Stability of States**   : Let states $Q$ of DFA $(\Sigma, Q, \delta, q_0, F)$ be embedded in the metric space of $\mathcal{R}^n$ and mapping $\delta : Q \times \Sigma \to Q$ be the transition function. Consider transitions for state $q \in Q$ s.t. $\delta(q, a) = q$ for some $a \in \Sigma$. By Theorem 3.2 and Definition 3.5, the error in stability of states can be estimated. The stability of states directly influences the stability of predictions.

Thus, we wish to ask the following question

**Research Question:**

1. Does there exist a class of stable RNNs, that, for any regular language over dataset or distribution generated over any probability distribution ($\mathbb{P}_d$) clusters the internal states in reasonable time to approximate minimal DFA with high probability ($\mathbb{P}_m$), such that difference between states of RNN ($\hat{Q}$) and DFA ($Q$) in euclidean space is within the small bound. $||Q - \hat{Q}||_2^2 < \epsilon$

2. RNNs that are partially trained and do not achieve perfect accuracy (i.e. $100\%$) are known as partially trained RNNs. We consider them as a stochastic state estimator for that language. How do different DFA extraction methods affect optimal state estimation from partially trained RNN such that the extracted state ($\hat{Q}_e$) is much closer to the optimal state ($\hat{Q}$) of stable RNN. $||\hat{Q} - \hat{Q}_e||_2^2 < \epsilon_e$, where $\epsilon_e \geq \epsilon$

## 4 DATA AND MODELS

In all our experiments, RNNs (both first and second order) are trained on a dataset of positive and negative examples of strings. These strings are randomly generated with varied lengths for any given formal language. Below, we explain our dataset creation procedure and training strategies.

**Tomita Grammars**: One of the widely used datasets for benchmarking grammatical inference algorithms is based on 7 Tomita grammars (definitions are given in Table 1) Tomita (1982). In this work, to have a fair comparison with the literature we adopt the dataset creation script provided by (Bhattamishra et al., 2020a). However (Bhattamishra et al., 2020a) do not provide **separate validation set** for model selection and hyperparameter optimization. This approach of optimizing models on a test set creates biased prediction. Thus to overcome this issue we use their bin_0 test set as the validation set and generated our own test bin_0 of 2000 samples for each of the seven Tomita grammars. The train, validation, and test bin_0 sets have a minimum string length of 2 and a maximum 50. The test bin_1 has strings of length $\in [51, 100]$. The train sets have 10000 samples each, while validation and test sets have 2000 samples each. We ensured that there was no overlapping of samples in any sets, to avoid any potential data leakage.

**Dyck Grammars**: A Dyck word is a set of balanced parenthesis. A Dyck grammar can be written as a context-free grammar with production rule $S \to "("S")"|\epsilon$ where "(" and ")" are left and right parenthesis respectively and $S$ is a non-terminal. Note that the string generated from such a rule is well-balanced.

Let $s_1, s_2 \in D$ be strings of dyck language $D$, then $s_1 + s_2 \in D$, where $+$ is a string concatenation operator. A string made up of $n$ pairs of well-nested parenthesis belongs to the Dyck-$n$ language.

We follow a split similar to Tomita datasets for train, validation, and test sets for Dyck-2,3,6, and 8. For training sets, we generate $20,000$ samples for each dyck language, while for test and validation sets, we generate $4000$ samples each. Tables 3 encapsulate the details.

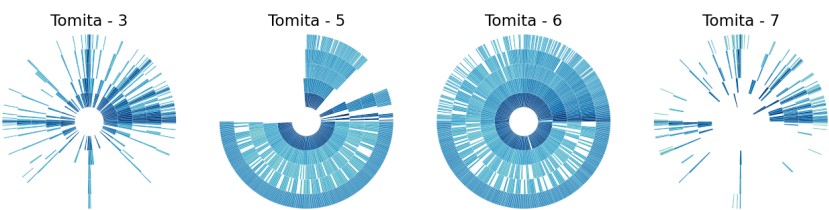

Figure 1: Distribution of strings of Tomita grammar in lexicographical order

**Partial Training:** RNNs when trained on regular languages, often achieve perfect accuracy on validation split. However, in the real world and even while dealing with natural language, it is difficult for NNs to achieve perfect accuracy. *We argue that an ideal extraction approach should even function in scenarios where the model doesn't achieve the perfect score.* Thus, to simulate these settings

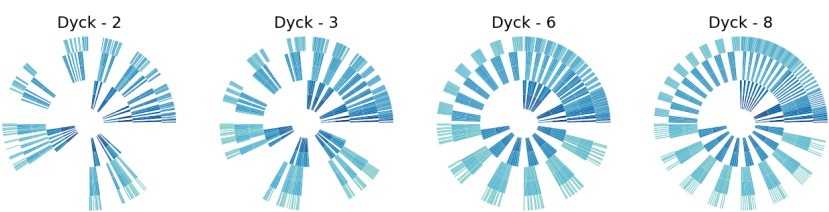

Figure 2: Distribution of strings of Dyck grammar in lexicographical order

and evaluate the stability of DFA extraction from a reasonably trained network, we create a set of partially trained neural networks on Tomita grammars. We stop the training of these networks as the validation accuracy $> 85\%$. However, some grammars like Tomita 1 and 2 often see higher accuracy as validation accuracy jumps around $69\%$ to $90\%$ between consecutive iterations. However, no network in this set achieves more than $90\%$ accuracy on validation and test sets. Other training details are provided in appendix C

## 5 RESULTS AND DISCUSSION

In this section, we will analyze the stability of DFA extraction of various grammars, respectively. Additional results are provided in appendix D.

**Tomita Grammars**    As discussed earlier, we trained four different RNNs on 7 Tomita grammar. In this section, we will primarily compare results from Tomita 3, 5, 6, and 7. As based on concentric ring representation, these grammars are slightly more complex compared to others. In the appendix D, we show model performances on Tomita 1, 2, and 4, respectively.

**Performance of RNNs on Tomita Grammars**    In our experiments, we train networks with state dimensions of size $n.s$, where $s$ is a multiplier that we vary in $[1, 2, 3, 4, 5]$ and $n$ is the minimal number of states in the ground truth DFA of the grammar, as given in the seminal work Tomita (1982). Tomita 6 grammar has a minimal state dimension of 3 (ref: table 2). From the concentric ring representation and analyzing the lexicographic space of grammar 6 (figure 1, we can see grammar 6 has an evenly distributed pair of positive and negative strings. Thus, making Tomita 6 a fairly complicated regular grammar, prior works have also noticed similar behavior Weiss et al. (2018); Wang & Niepert (2019). However, this provides us with an opportunity to examine better and compare the performance of $1^{st}$ and $2^{nd}$ order RNNs. This is evident from figure 13 where we can see that O2RNN consistently performs better than other RNNs representing higher learning capacity of $2^{nd}$ order recurrent cell operations. This is interesting to observe as Merrill (2019) has theoretically shown that LSTM is strictly more powerful than regular languages. We empirically validate the theoretical results in O2RNN that argue stability Mali et al. (2023a), as it is evident from figure 13, which shows the mean accuracy of RNNs across 10 seeds, where O2RNN constantly achieves higher accuracy across state-dimension. In figure 15, we also report the standard deviation of all the models across seeds. This is an important measure to provide statistical significance of any model. For simplicity, we show the standard deviation of models with a state dimension of $2n$, where $n$ is the number of states in the minimal ground truth DFA. However, these results do extrapolate to other state dimensions, and we observe consistent trends across settings.

As we have observed above, with a special emphasis on Tomita-6, *O2RNN is more stable* as it gives **0%** standard deviation in its performance as compared to **15.81%**, **23.85%** and **26.35%** deviation of LSTM, GRU and MIRNN respectively across 10 seeds on test bin 0. Similar performance was on test bin 1. Given that O2RNN can insert Giles & Omlin (1993) and refine Giles et al. (1992) rules and often learn with fewer data, we believe these results are an important direction towards building neuro-symbolic AI systems.

**Comparison of DFA extracted from Tomita Grammars**    DFAs extracted using quantization and equivalence query methods show similar accuracies on Tomita languages. In figure 10 we report the mean performance of DFA's extracted by quantization approach using self-organizing maps across

10 seeds. Similarly, figure 11 shows the mean performance of DFAs extracted using $L^*$. Figure 12 shows the standard deviation of performance of DFA's from mean accuracy across 10 seeds. Here, we can observe that both methods show similar stability in terms of the performance of DFAs on the test set (bin-0). Even though the performance of DFAs extracted by the two methods is similar for Tomita languages, there is a noticeable difference in the number of states in the extracted DFAs. From figure 3 we can observe that the deviation in the number of states extracted from the mode number of states across 10 seeds is quite large for DFAs extracted using $L^*$. *For example, we can observe from fig 3, in Tomita-5, the maximum number of states extracted by $L^*$ **is above 200** for LSTM, GRU, and O2RNN and about* 150 *for MIRNN*. Note that the number of states in Tomita-5 ground truth DFA is only 4. However, the number of states extracted by quantization methods consistently remains low ( **under 10 states** for the majority of seeds ).

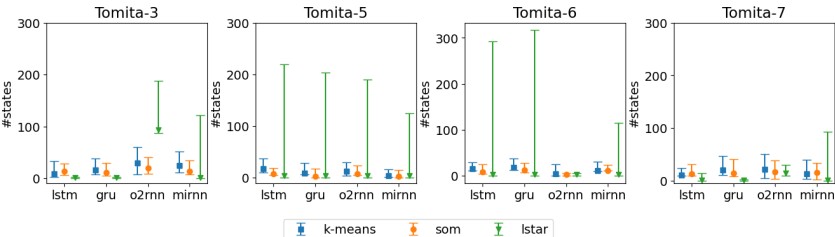

Figure 3: Min, max, and mode of the number of states extracted from $1^{st}$ and $2^{nd}$ order RNNs trained on Tomita grammars

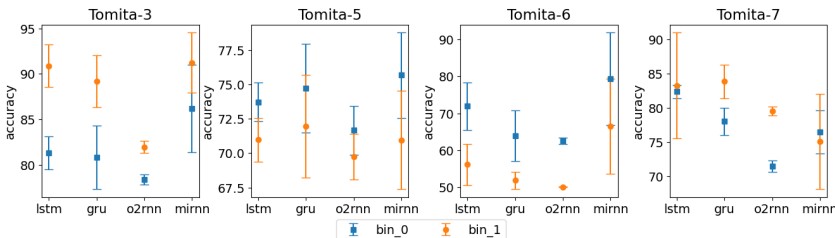

Figure 4: Mean Accuracy and Standard Deviation of $1^{st}$ and $2^{nd}$ order Recurrent Neural Networks with restricted training on Tomita languages. The training was stopped when validation accuracy crossed 85%.

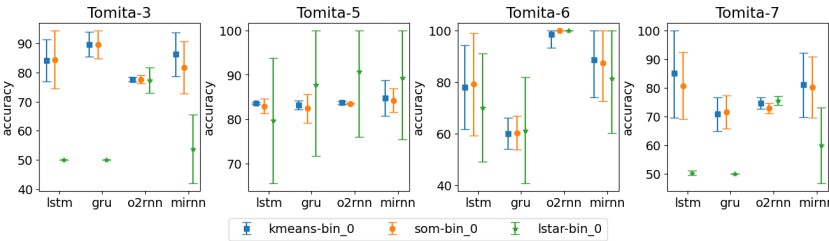

Figure 5: Mean and Standard Deviation of the accuracy of DFAs extracted from $1^{st}$ and $2^{nd}$ order RNNs with restricted training (85% validation accuracy) on Tomita languages.

**Partially Trained Tomita Grammars**   As argued in the main paper, the DFA extraction from partially trained RNNs might provide a better insight into the stability of the two extraction methods. Figure 4 shows the performance of partially trained RNNs on Test bin_0 and bin_1, and figure 5 shows the performance of DFAs extracted from partially trained RNNs. Here, we can observe that $L^*$ fails to extract valid DFA for LSTM and GRU for Tomita-3 and Tomita-7 and also shows high instability for Tomita-5 and Tomita-6. In contrast, K-means and SOM-based extraction produces relatively stable DFAs. Also, we can observe that DFAs extracted from O2RNN by quantization methods are

more stable as they show low deviation and better mean performance across 10 seeds. Note that in Tomita-3 and Tomita-7, even though the mean performance of DFA extracted from LSTM is higher, it also has higher instability than O2RNN. *In Tomita-6, which is a relatively difficult grammar to learn, DFA extracted from O2RNN has the highest mean accuracy and the lowest deviation.* It is important to note that none of the prior work conducted this type of study with partial strings. From the result, it is evident that even in scenarios where the model exhibits lower generalization performance, quantization-based methods can still extract stable rules, which is an important directional towards building responsible AI systems.

**Dyck Languages**    In this section, we analyze the stability of DFA extraction methods on dyck language.

**Performance of RNNs on Dyck Languages**    For all RNN variants, figure 22 shows that the mean accuracy increases with an increase in state dimension before stabilizing, indicating that for complex grammars, more parameters are required. We observe that LSTM and GRU can easily reach near $100\%$ accuracy with a lesser number of neurons in the hidden state compared to O2RNN and MIRNN. Figure 23 shows the standard deviation of the accuracy of RNNs on Dyck languages across 10 seed. Similar to Tomita, we evaluate all models with hidden state dimension $= 2k$ where $k$ is the type of Dyck language, i.e. Dyck-$k$ . We observe all networks behave similarly, whereas GRU shows highly better performance. However, GRU and LSTM are more sensitive to string lengths as the standard deviation is higher for test bin 1, which has strings of length 51-100 as compared to test bin 0 with string lengths 2-50. O2RNN and MIRNN show similar standard deviations for both test sets. *Thus indicating higher-order networks better understand the structure of the language as opposed to memorization.*

**Performance comparison of DFA extracted from Dyck Grammars**    As highlighted earlier, GRU gets higher accuracy compared to other RNNs in recognizing Dyck languages. However, from figure 6, we can see that DFA extracted from O2RNN using the quantization method consistently outperforms other models DFAs over a broader range of hidden state dimensions. We can also see this phenomenon for DFAs extracted by $L^*$ (fig 7) for Dyck-6 and Dyck-8 languages too. Figures 6 and 8 show the ability of O2RNN to understand and encode rules most stably, which can be extracted effectively by using quantization methods. *This shows that O2RNN is trying to learn the underlying grammatical structure as opposed to memorizing the language.*

Furthermore, by comparing figures 6 and 7, we observe that DFAs extracted using quantization methods consistently outperform DFAs extracted using $L^*$. This issue becomes even more evident by analyzing Figure 8, which shows the stability issues of $L^*$ based approaches while extracting DFA from Dyck languages. On multiple occasions, DFAs extracted by $L^*$ even fail to perform better than *random guess* ($50\%$). We can also observe the instability of $L^*$ in the number of states in the extracted DFAs as we vary the state dimensions(figure 24) and even across 10 seeds ( figure 9. The number of states in DFA's extracted by $L^*$ can range in the order of **100** states while quantization methods consistently produce DFA's with less than **50** states.

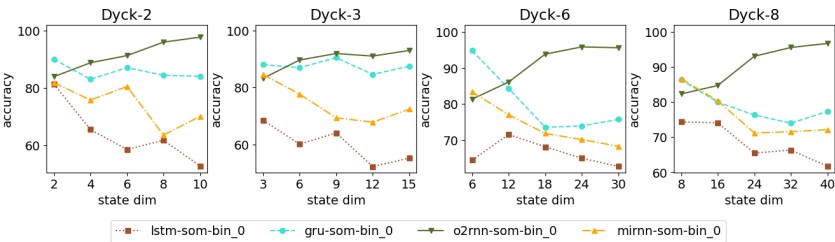

Figure 6: Mean accuracy of DFA extracted from RNNs by quantization methods across 10 seeds on Dyck grammars

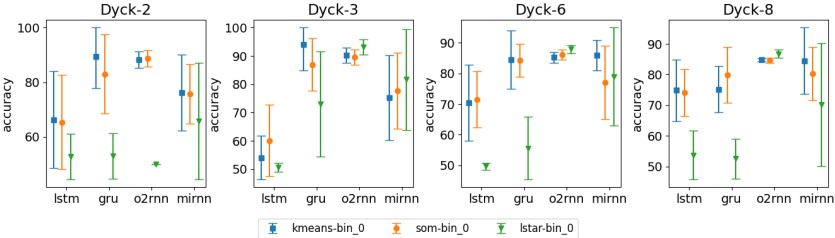

Figure 7: Mean accuracy of DFA extracted from RNNs by $L^*$ across 10 seeds on Dyck grammars

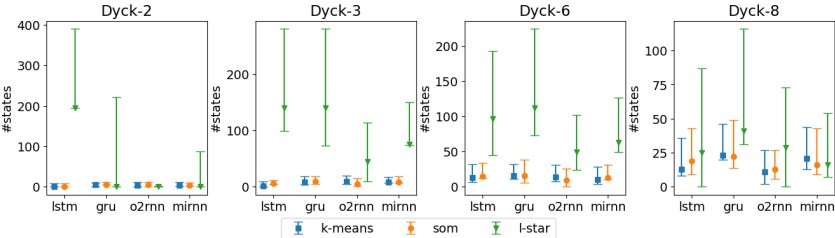

Figure 8: Mean and Standard deviation of accuracy of DFA extracted from $1^{st}$ and $2^{nd}$ order RNNs on Dyck grammars.

Figure 9: Min, max and mode of the number of states extracted from $1^{st}$ and $2^{nd}$ order RNNs on Dyck grammars.

## CONCLUSION

We have studied how various classes of RNNs (3600 models) trained to recognize regular formal languages stably represent knowledge in their hidden state. Specifically, we have asked how stable and close these internal representations are compared to Oracle state machines (**R1**) and whether we can stably extract minimal deterministic state machines across various settings, even in scenarios where the network is partially trained (**R2**). To prove our hypothesis, we conducted extensive experiments and show that the state machine extracted from the $2^{nd}$ RNN network (O2RNN) is much more stable and achieves $0\%$ standard deviation, followed by LSTM with $4\% - 7\%$ when trained partially on Tomita languages. O2RNN is observed to be more stable on Dyck languages as well. Furthermore, we also showed that the equivalence query-based approach, $L^*$, shows significant instability in DFA extraction. Moreover, we also show even on a partially trained network, the quantization-based approach constantly outperforms $L^*$, indicating a promising direction on the importance of the quantization-based DFA extraction approach coupled with tensor connections.

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

## A  TOMITA GRAMMARS

| # | Definition |
|---|---|
| 1 | $a^*$ |
| 2 | $(ab)^*$ |
| 3 | Odd $[\![....]\!]_a$ of $a$'s must be followed by even $[\![....]\!]_b$ of $b$'s |
| 4 | All strings without the trigram $aaa$ |
| 5 | Strings $I$ where $[\![....]\!]_a(I)$ and $[\![....]\!]_b(I)$ are even |
| 6 | Strings $I$ where $[\![....]\!]_a(I) \equiv_3 [\![....]\!]_b(I)$ |
| 7 | $b^*a^*b^*a^*$ |

Table 1: Definitions of the Tomita languages. Let $[\![....]\!]_\sigma(I)$ denote the number of occurrences of symbol $\sigma$ in string $I$. Let $\equiv_3$ denote equivalence mod 3 (Mali et al. (2023a)).

## B  $L^*$ ALGORITHM

**L**$^*$ algorithm, proposed by ANGLUIN (1987) learns an unknown regular set $U$ over a fixed known finite alphabet $A$ from a minimally adequate Teacher $T$. $U$ is represented as a DFA. A minimally adequate Teacher with knowledge of regular set $S$ is any system that can answer two types of queries:

1. Membership Query: For a given Teacher $T$ and a membership query $Q_M$ for string $s$ the teacher responds as :

$$Q_M(T, s) = \begin{cases} 1 & s \in S \\ 0 & \text{otherwise} \end{cases}$$

2. Equivalence Query: The Teacher $T$ responds to an equivalence query $Q_C$ as:

$$Q_C(T, U) = \begin{cases} 1 & \text{if } S \subseteq U \text{ and } U \subseteq S \\ t & t \in \ S \cup U - S \cap U \end{cases}$$

$L^*$ maintains an observation table as a Hankel Matrix ($H(p.e) \to \{0, 1\}$) where $p \in P \cup P.A$, where $P$ is a finite prefix closed set. $b \in E$, where $E$ is a finite suffix closed set. $L^*$ performs a sequence of $Q_M$ and $Q_C$ with the given teacher $T$ and concludes when $H$ is closed and consistent. $H$ is closed iff $\forall t \in P.A \ \exists \ p' \in P$ s.t. row($p'$) = row($t$) and $H$ is consistent iff for $p_1, p_2 \in P$ and row($p_1$) = row($p_2$), then $\forall a \in A$ row($p_1.a$) = row($p_2.a$). Here (.) is the string concatenation operator. A DFA $(Q, \Sigma, \delta, q_0, F)$ is then constructed from $H$ with $\Sigma = A$, $Q = \{row(p) : \forall \ p \in P\}$, $q_0 = row(\epsilon)$, $F = row(p) : p \in P$ and $H(p) = 1$, and $\delta(row(p), a) = row(p.a)$

In the absence of a minimally adequate teacher $T$, a stochastic Oracle $O$ capable of answering $Q_M$ and $Q_C$ can also be used. We follow the approach of Weiss et al. (2018) and use the RNNs as oracle to extract DFAs for each of the 3200 models. One observation we made here is that the extraction heavily depends on initial counter-examples. This does not prove good for Dyck grammars especially.

## C  TRAINING DETAILS

The weights of all 4 RNNs are initialized using fan-in/fan-out initialization; biases are initialized to 1. The batch size is set to 2048, and all networks are optimized using stochastic gradient descent with an initial learning rate of 0.01 for a maximum of 15000 iterations with a single-cycle learning rate scheduler (Smith & Topin, 2019). We also use early stopping with patience criteria of 1000 iterations on the validation set. All models are trained using binary-cross entropy. We also introduce a data augmentation approach that helps networks with fewer parameters to converge on complex grammars such as Tomita-3 and Tomita-6. At each step, we classify partial string and calculate the loss at the time stamp.

Prior works focused on recognizing grammatical sequences Bhattamishra et al. (2020a); Mali et al. (2021a); Suzgun et al. (2019a) have shown that neural networks with few parameters can effectively learn Tomita and Dyck languages. However, works focused on rule extraction (Weiss et al., 2018) have used large networks with hidden sizes ($> 50$) and layers ($> 1$). This doesn't align with

theoretical results, which state that a single-layer network can recognize regular grammars Merrill (2019); Mali et al. (2023a). In some cases, it is difficult to evaluate the memorization effect vs the generalization capability of the model if the number of parameters is more than the samples. Thus, to better understand the true generalization capability of models, we train our networks on a small range of hidden state sizes. For Tomita grammars, we use hidden sizes, which are multiples of the number of states in the ground truth DFA. Table 2 provides the number of states of minimal DFA designed to recognize 7 Tomita grammars. For Dyck-$k$ languages, we use multiples of $k$ as hidden size. Furthermore, we also compare the stability of extraction of DFAs on a hidden state size multiple of 2, but we also train our models on multiples of 1, 3, 4, and 5. The results of extended multiples are covered in the ablation study and supplementary material. In Table 4, we report the number of parameters for the smallest and the largest models used to recognize each language. In our experiments, the smallest model had only **21** parameters, whereas the largest model had **3126**.

|  | Tomita Grammars | | | | | | |
| --- | --- | --- | --- | --- | --- | --- | --- |
|  | 1 | 2 | 3 | 4 | 5 | 6 | 7 |
| #states | 2 | 3 | 5 | 4 | 4 | 3 | 5 |

Table 2: Minimal number of states in the ground truth DFA of Tomita grammars

**Computational budget:** As notified in prior sections we train 4 types of RNN models for 7 Tomita grammars and 2 dyck grammars, with 5 configurations of hidden state sizes, across 10 initial seeds. Additionally, we have partially trained RNNs with all the above configurations for 7 Tomita grammars, resulting in a total 3200 neural networks. Each network takes, on average, approximately 30 mins to train on a single Nvidia 2080ti GPU. Thus, we spend a total of 1600 GPU hours for training. Testing and DFA extraction required additional GPU hours.

# D    ADDITIONAL RESULTS

|  | String Length [min - max] | Tomita [1-7] | Dyck [2,3,6,8] |
| --- | --- | --- | --- |
| Train | 2-50 | 10000 | 20000 |
| Val | 2-50 | 2000 | 4000 |
| Test Bin 0 | 2-50 | 2000 | 4000 |
| Test Bin 1 | 51-100 | 2000 | 4000 |

Table 3: Number of samples for each grammar in the datasets

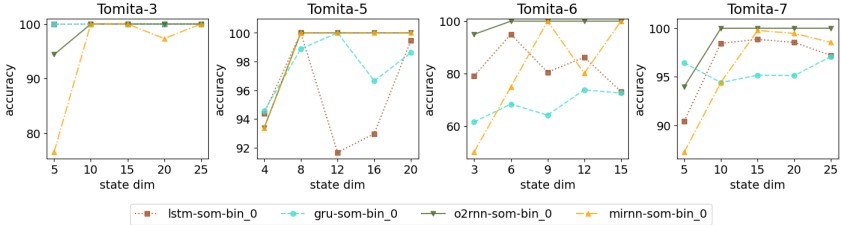

Figure 10: Mean accuracy of DFAs extracted from various RNNs using quantization method (clustering by SOM).

| grammar | state dim | lstm | gru | o2rnn | mirnn |
|---------|-----------|------|-----|-------|-------|
| tomita_1 | 2 | 67 | 51 | 21 | 23 |
|          | 10 | 651 | 491 | 421 | 191 |
| tomita_2 | 3 | 112 | 85 | 43 | 37 |
|          | 15 | 1276 | 961 | 931 | 361 |
| tomita_3 | 5 | 226 | 171 | 111 | 71 |
|          | 25 | 3126 | 2351 | 2551 | 851 |
| tomita_4 | 4 | 165 | 125 | 73 | 53 |
|          | 20 | 2101 | 1581 | 1641 | 581 |
| tomita_5 | 4 | 165 | 125 | 73 | 53 |
|          | 20 | 2101 | 1581 | 1641 | 581 |
| tomita_6 | 3 | 112 | 85 | 43 | 37 |
|          | 15 | 1276 | 961 | 931 | 361 |
| tomita_7 | 5 | 226 | 171 | 111 | 71 |
|          | 25 | 3126 | 2351 | 2551 | 851 |
| dyck_2 | 2 | 91 | 69 | 33 | 29 |
|        | 10 | 771 | 581 | 721 | 221 |
| dyck_3 | 3 | 172 | 130 | 88 | 52 |
|        | 15 | 1576 | 1186 | 2056 | 436 |
| dyck_6 | 5 | 559 | 421 | 553 | 157 |
|        | 30 | 5671 | 4261 | 13561 | 1501 |
| dyck_8 | 8 | 937 | 705 | 1233 | 257 |
|        | 40 | 9801 | 7361 | 30481 | 2561 |

Table 4: Number of parameters in RNNs for smallest and largest hidden state size used for each grammar

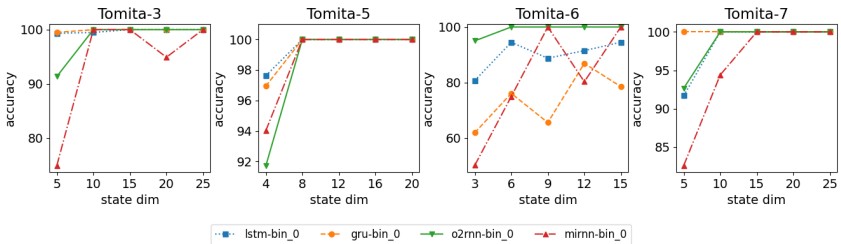

Figure 11: Mean accuracy of DFAs extracted from various RNNs using equivalence query method $(L^*)$

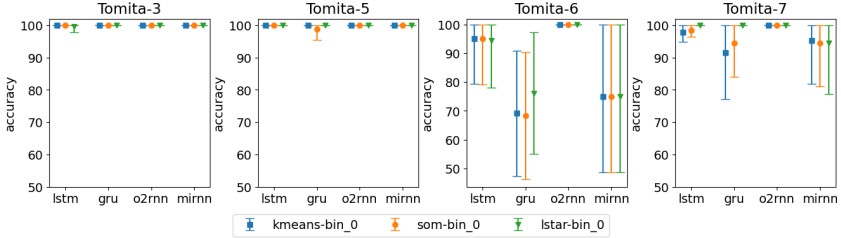

Figure 12: Mean and standard deviation of accuracy of DFAs extracted from $1^{st}$ and $2^{nd}$ order RNNs trained on Tomita grammars over 10 seeds

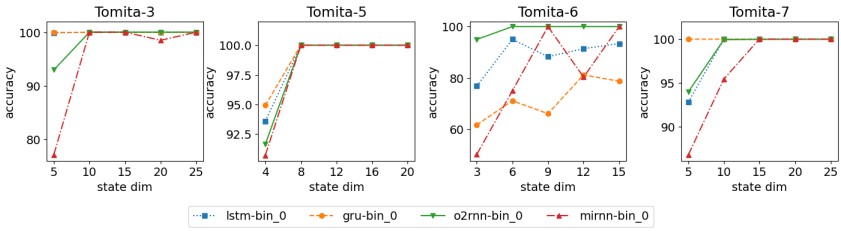

Figure 13: Mean Accuracy of $1^{st}$ and $2^{nd}$ order Recurrent Neural Networks on Tomita 3, 5, 6 and 7

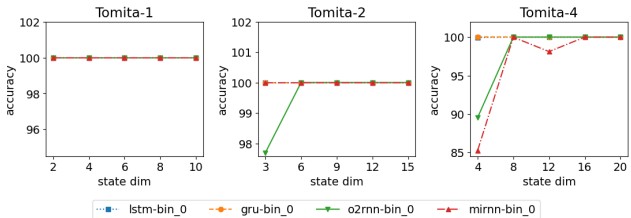

Figure 14: Mean Accuracy of $1^{st}$ and $2^{nd}$ order Recurrent Neural Networks on Tomita 1, 2 and 4

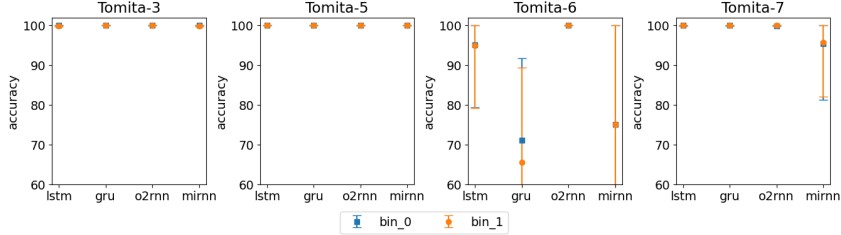

Figure 15: Mean and Standard Deviation of $1^{st}$ and $2^{nd}$ order Recurrent Neural Networks trained on 3, 5, 6 and 7.

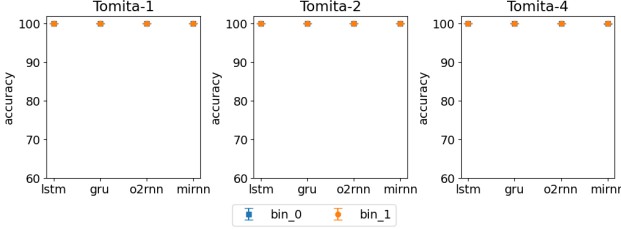

Figure 16: Mean and Standard Deviation of $1^{st}$ and $2^{nd}$ order Recurrent Neural Networks trained on Tomita 1, 2 and 4.

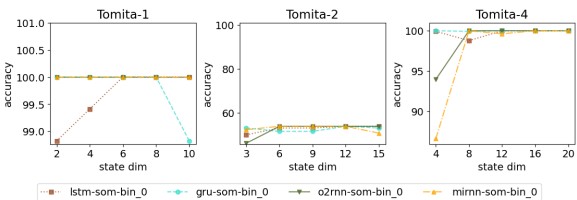

Figure 17: Mean accuracy of DFA extracted from $1^{st}$ and $2^{nd}$ order Recurrent Neural Networks trained on Tomita 1, 2 and 4 by quantization methods

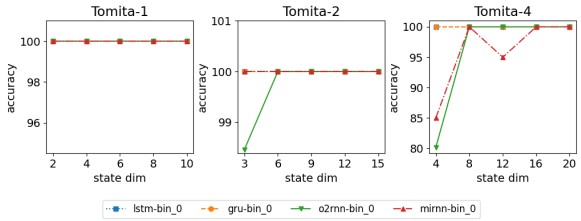

Figure 18: Mean accuracy of DFA extracted from $1^{st}$ and $2^{nd}$ order Recurrent Neural Networks trained on Tomita 1, 2 and 4 by $L^*$

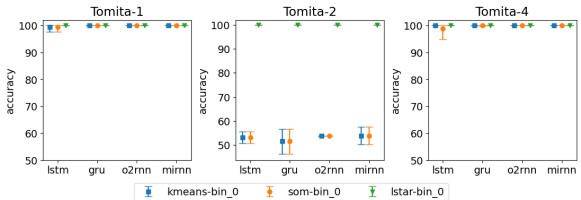

Figure 19: Mean and Standard Deviation of DFA extracted from of $1^{st}$ and $2^{nd}$ order Recurrent Neural Networks trained on Tomita 1, 2 and 4.

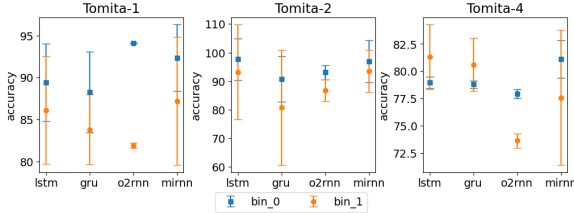

Figure 20: Mean and Standard Deviation of $1^{st}$ and $2^{nd}$ order Recurrent Neural Networks partially trained on Tomita 1, 2 and 4.

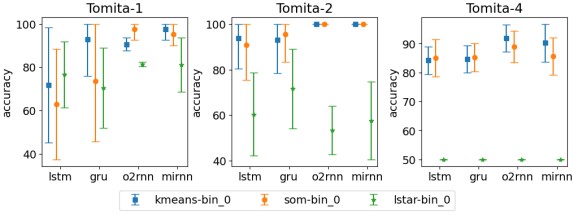

Figure 21: Mean and Standard Deviation of DFA extracted from $1^{st}$ and $2^{nd}$ order Recurrent Neural Networks partially trained on Tomita 1, 2 and 4.

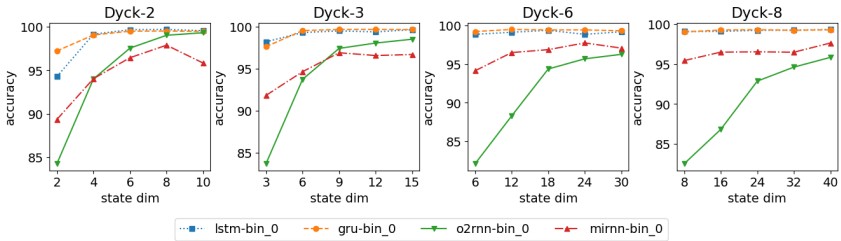

Figure 22: Mean Accuracy of $1^{st}$ and $2^{nd}$ order Recurrent Neural Networks on Dyck-2, 3, 6 and 8 languages

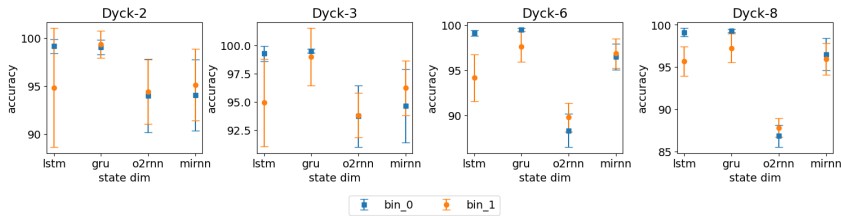

Figure 23: Mean Accuracy and Standard Deviation of $1^{st}$ and $2^{nd}$ order Recurrent Neural Networks trained on Dyck grammars

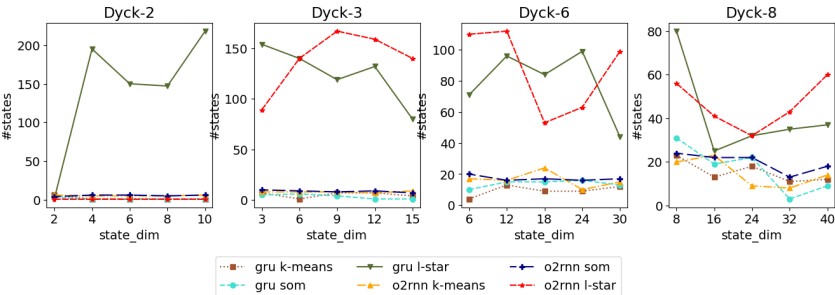

Figure 24: Mode of the number of states extracted from $1^{st}$ and $2^{nd}$ order RNNs on Dyck grammars across 10 seeds.

