# OpenReview forum: "Stability Analysis of Various Symbolic Rule Extraction Methods from Recurrent Neural Network"
_ICLR.cc/2024/Conference — Submitted to ICLR 2024_

### Official Review · Reviewer_AfSv · 2023-10-30

**Soundness:** 3 good
**Presentation:** 2 fair
**Contribution:** 3 good
**Rating:** 8
**Confidence:** 3

**Summary:**

This paper examines the efficiency and stability of various RNN architectures in the task of DFA extraction. It also compares the stability of two rule extraction methods, quantization and equivalence query, applied to trained RNNs. The findings indicate that the quantization-based approach is both more stable and generally outperforms the equivalence query method. Additionally, the study reveals that the DFA extracted using O2RNN surpasses other RNN architectures in terms of performance and stability.

**Strengths:**

To the best of my knowledge, this is the first paper that delves into the stability of rule extraction methods from trained RNNs. The paper is well-structured and straightforward. Moreover, the experiments conducted are thorough and systematic, and the results are indeed potentially useful in practice. I believe this work will be a solid contribution to the community.

**Weaknesses:**

I think the mathematical formulation of the problems considered in this paper can be more precise and more detailed, to make it easier to understand.

**Questions:**

No.

---

> ### Author Response · Authors · 2023-11-19
>
> Dear Reviewer AfSv,
>
> We thank you for your insightful review and comments.
>
> With the help of your feedback, we have updated our paper with mathematical formulations for the following sections:
> 1. Stability Analysis
> 2. Formulation of R1 and R2 founded in the mathematics of stability
> 3. Updated mathematical formulation of L* algorithm (in Appendix)
> 4. Rephrased our conclusion to show that results are well in the mathematical formulations of R1 and R2, and our observations have theoretical backing.
>
> **All updates in the paper are reflected in blue font color**
>
> We have also provided a paper overview with mathematical formulations of **R1** and **R2** in the comments above.
>
> Thank You !

---

> > ### Author Response · Authors · 2023-11-29
> > **A Gentle Reminder**
> >
> > Dear Reviewer Afsv,
> >
> > Thank you for your positive review and valuable insights on our work. We have addressed all your comments/concerns in our detailed response and believe we have resolved all problems. Should any issues remain, we are ready to provide additional clarifications.
> > As the rebuttal phase is nearing its deadline, we look forward to engaging in a timely discussion.
> > Thank you again for your time and effort!
> >
> > Best regards,
> > Authors

---

### Official Review · Reviewer_nZtd · 2023-11-10

**Soundness:** 4 excellent
**Presentation:** 4 excellent
**Contribution:** 4 excellent
**Rating:** 5
**Confidence:** 1

**Summary:**

This paper conducts an empirical analysis of two competing methodologies for rule extraction—quantization and equivalence query—with the goal of elucidating the internal mechanisms of diverse RNN architectures. The authors identify specific scenarios where each strategy proves effective, and they substantiate their findings through comprehensive experiments designed to validate the presented conclusions.

**Strengths:**

The paper is well presented, by providing substantial background information and experiments.

**Weaknesses:**

See questions.

**Questions:**

Honestly, my familiarity with current RNN research, including rule extraction strategies, is limited. Consequently, I find it inappropriate to assess the topic, method, or the novelty of the work. However, I do have several questions for discussion.

- It would enhance the paper if the authors could furnish additional insights or intuition regarding the significance of exploring the stability of rule extraction strategies. Alternatively, a clearer definition of stability in this context would also be helpful. In the current work, stability appears to be synonymous with the variance of performance.

- A more thorough justification for the conclusion would be beneficial. Instead of solely benchmarking several methods across various settings, delving into the rationale behind the conclusions drawn would strengthen the paper and provide a more comprehensive perspective.

---

> ### Author Response · Authors · 2023-11-19
>
> Dear Reviewer nZtd,
>
> We thank you for your insightful reviews and comments.
>
> We base our notion of stability of DFA extraction on Brouwers's Fixed Point Theorem and build on previous theoretical works on the stability of state estimation by RNNs.
>
> ---
> **Theorem 1**
> BROUWER’S FIXED POINT THEOREM: For any continuous mapping $f:Z \rightarrow Z$, where Z is a compact, non-empty convex set, $\exists \ z_f$ s.t. $f(z_f) \rightarrow z_f$
>
> **Definition 1**
> Let $\mathcal{E}$ be an estimator of fixed point $z_f$ for mapping $f:Z \rightarrow Z$, where $Z$ is a metric space. $\mathcal{E}$ is a stable estimator iff estimated fixed point $\hat z_f$ is in immediate vicinity of $z_f$, i.e. $ | \hat z_f - z_f | < \epsilon, $ for arbitrarily small $\epsilon$.
>
> **Corollary 1**
> Stability of estimator $\mathcal{E}$ can be shown by iteratively computing $z^{(t+1)} = \mathcal{E_f} (z^{(t)})$ with $z^{(0)}$ in the neighborhood of $z_f$. For stable estimator $\mathcal{E}$, $\lim_{t\rightarrow \infty}z^{(t)}  = z_f$. Neighborhood of $z_f$, $N_{z_f}$ is a set of points $z_n \in N_{z_f}$ near $z_f$ s.t. $|z_n - z_f| < \delta $.
>
> Omlin and Giles, 1996,  use this idea of stability to show that for provably stable second-order RNNs, the sigmoidal discriminant function should drive neuron values to near saturation. Similarly, Stogin and Mali, 2020 use the fixed point analysis to prove the stability of differentiable stack and differentiable tape with RNNs.
>
> **Definition 2**
> Let $\mathcal{E^*}$ be a stochastic fixed point estimator for $f$. The estimated $RMS$ error for fixed point estimation  $\hat z_{f;i} = \mathcal{E^*}(f)$ can be defined as:
> $$
>     e_{rms} = \sqrt{\frac{1}{n}\sum_{i=0}^{n} || \hat z_{f;i} - z_f ||^2}
> $$
>
> Note that when true $z_f$ is unknown, the law of large numbers allows us to use the estimated mean $\overline{z_f}$ instead. Also the error estimate $e_{rms}$ is the standard deviation from $z_f$
>
> **Stability of States**
> Let states $Q$ of DFA ($\Sigma, Q, \delta, q_0, F$) be embedded in the metric space of $\mathcal{R}^n$ and mapping $\delta : Q \times \Sigma \rightarrow Q$ be the transition function. Consider transitions for state $q \in Q$ s.t. $\delta(q, a) = q$ for some $a \in \Sigma$. By Theorem 1 and Definition 2, the error in the stability of states can be estimated. The stability of states directly influences the stability of predictions.
>
> ---
>
> **We have updated our paper with this stability analysis and updated the Conclusion section to reflect that. All changes made in the paper are highlighted in blue font color**
>
> ---
>
> With the help of your feedback, we have grounded our motivation for this paper into two research questions, R1 and R2, as mentioned in the Paper Overview commented above, and updated our Conclusion accordingly. In the conclusion section, we show that the empirical results for DFA stability are well within the theoretical framework of stable state estimation and answers R1 and R2.
>
> Thank You !

---

> > ### Author Response · Authors · 2023-11-29
> > **A Gentle Reminder**
> >
> > Dear Reviewer nZtd,
> >
> > Thank you for your positive review and valuable insights on our work. We have addressed all your comments/concerns in our detailed response and believe we have resolved all problems. Should any issues remain, we are ready to provide additional clarifications.
> > As the rebuttal phase is nearing its deadline, we look forward to engaging in a timely discussion.
> > Thank you again for your time and effort!
> >
> > Best regards,
> > Authors

---

### Official Review · Reviewer_17Qn · 2023-11-22

**Soundness:** 3 good
**Presentation:** 1 poor
**Contribution:** 2 fair
**Rating:** 5
**Confidence:** 3

**Summary:**

This paper is an empirical study on the capability and stability of rule extraction of deterministic finite automata (DFA) of multiple regular languages by a number of RNNs. Results show that second-order RNNs like O2RNN have better stability in extracting DFAs than first-order ones like GRUs and LSTMs, validating prior theoretical work. The results also show that quantization-based approaches extract fewer states than equivalence query methods. In addition, this paper also contains many comparison analyses for the effects of different architecture hyperparams on rule extractions.

**Strengths:**

This paper establishes its conclusion through solid empirical investigations. By varying rule-extraction methods, languages, architectures, and a bunch of hyper-parameters, this paper gives some valuable information on RNNs' performance in these types of tasks, and how consistent they are. These results could in turn give some valuable feedback to the theoretical community, where the motivation of these questions is from.

A possible potential for the approach investigated in this work is that it could serve as inspiration for the design of tasks to understand the capability of other types of neural networks that could potentially receive context-free language as inputs (say when context-free language is used to evaluate language models' ability).

**Weaknesses:**

1. The way this paper motivates the reader, and how the background is introduced, is somewhat too narrowly focused such that only researchers working in the most vicinity of the DFA extraction area could appreciate it.
2. Even though this paper aims to provide empirical validation of some prior theoretical results, its exposition of the theoretical background is a little incomplete and not well organized (for example, what is  $\mathcal{E}_f$ and what's its relation with $\mathcal{E}$ and $\mathcal{E}^*(f)$? What does the subscript $|_i$ means in notation $\mathcal{E}^*(f)|_i$?) I think the paper would benefit hugely from a rewriting of these sections, to fill the gap between rough discussions of prior works and actual mathematical backgrounds and the question this paper is tackling.
3. The importance of the key evaluation metrics is not well motivated, and corresponding results are not well explained. For example, this paper should motivate more on why the number of extracted states can be used to evaluate DFA extraction strategies, and why lower numbers are preferred (yes the conclusion section contains better forms of your research questions but still why aren't they presented earlier?).
4. The overall significance of the results in this paper is limited by the paper's choice to study only RNNs and rule extraction approaches, as they do not imply any further empirical messages, as the paper has not pointed to any other potential implications.

**Questions:**

See above weaknesses.

---

> ### Author Response · Authors · 2023-11-28
>
> (Part 1/3)
>
> Dear Reviewer 17Qn,
>
> We thank you for the review and really appreciate your valuable feedback. We have made some changes to the notation as suggested by you. Please allow us to provide some clarifications for your questions:
>
> 1. We understand that the community of researchers working on DFA extraction is relatively small. However, this methodology is well accepted in the community working towards explainable AI. Since the dawn of computing, DFAs have been a fundamental structure that has guided hardware and software designs. Chomsky’s hierarchy provides us with a framework that allows us to classify problems based on their complexity and design machines to solve them. In the case of neural networks, Delétang, et. al. 2022, have shown the performance of RNN, memory-augmented NNs, and Transformer networks on a variety of tasks classified into different categories according to Chomsky’s hierarchy. In neural networks, RNNs are the only networks that can model state transitions directly in their architecture. Other networks like Transformers are shown to solve stateful problems, but **state extraction from Transformers has remained an open research question**. On the contrary, there have been a lot of works that demonstrate the extraction of states from RNNs and create a DFA. However, RNNs are stochastic systems. Omlin and Giles, 1996,  have shown the fixed point stability of RNN can be achieved using constraints on discriminant function. However, ours is the first empirical study showing the stability of DFAs extracted from 1st and 2nd-order RNN architectures and contrasting the performance of quantization and equivalence query mechanisms. We show that 2nd-order RNNs are better at state estimation, and quantization methods can stably extract these states.
>
> **Why it is important**: Equivalence with DFA helps in the verification of NNs (Wang et al., 2020) and formal verification (Alur et al., 2009). Extracting automata from RNNs is termed interpretable by extraction (Marzouk et al., 2020). Thus, the model can be termed interpretable if it shows stable rule extraction behavior.  Third, O2RNN can refine and insert rules (Giles and Omlin, 93), an essential direction towards neuro-symbolic and XAI. Thus if we can demonstrate the stability of these networks, then we can use them in various high-risk domains. For instance, automata are used in network and cyber security, DNA, health, controls, etc. Thus, this study will open the door to other fields where explainability and verification are essential.
>
> **Why it is relevant to the ICLR audience**: We believe the ICLR community will greatly benefit from this empirical study and theoretical foundation of RNNs, showing the stability.
>
> **Why only RNN**: It is important to note despite the impressive performance of transformers-models, they struggle to recognize regular grammar (Bhattamishra et al., 2020, Mali et al., 2023), computationally even after the increasing number of parameters cannot recognize simple regular languages (Delétang et al., 2023), thus have low expressive capability and in long-range arena tasks there performance degrades (Orvieto et al., 2023).  Thus extracting state machines is challenging for transformers as they have low encoding capability and suffer from large variance in their prediction. Furthermore, RNNs have a faster inference time than transformers, so studying their stability is important.
>
> **Why not CFGs**: We have shown results with Dyck grammars, which is a CFG (section 5, figures 6-9). However, to show stability for all classes of CFGs is out of scope and we highlight why. As demonstrated theoretically by Merrill 2023 and empirically by Delétang et al., 2023, RNNs, including LSTMs, fail to recognize context-free languages. To recognize CFGs, one needs access to external memory (Tape, Stack, or Queue). These networks are known as memory-augmented RNNs. Memory-augmented NNs have two components, controller, which can be GRU/RNN/LSTM/02RNN, etc, and differentiable memory, thus introducing two forms of instability, one at the controller end and the other at the memory end. Furthermore, Joulin and Mikolov 2015, empirically showed that stack-RNN trained using Backprop through time suffers from instability; recently, Stogin et al., 2019 theoretically showed the stability of differentiable stacks. **With memory-augmented NNs, only quantization-based approaches are shown to extract CFGS (Das et al., 92)**. Thus this shows why studying the stability of extraction approaches is important and guides future studies focused on memory-augmented RNNs.

---

> ### Author Response · Authors · 2023-11-28
>
> (Part 2/3)
>
> 2. We greatly appreciate your suggestions on notations. Here is more clarification on the notations:
>
> + $\mathcal{E}$ is an arbitrary fixed point estimator
> + $\mathcal{E}_f$ is fixed point estimator for mapping $f : Z \rightarrow Z$
> + $\mathcal{E}^*$ is a stochastic arbitrary fixed point estimator
> + $\mathcal{E}^*(f)$ is a stochastic arbitrary fixed point estimator  for mapping $f : Z \rightarrow Z$. We aim to change it to $\mathcal{E}^*_f$ for clarity and consistency.
> + $z_f$ is the true fixed point for mapping $f$.
> + $\hat{z}_f$ is the estimated fixed point for mapping $f$.
> + Since fixed point estimation is done in an iterative way, $\hat{z_{f;i}}$ is the fixed point estimated in $i^{th}$ iteration. For clarity and consistency we aim to change this to  $\hat{z}_{f}^{(i)}$
> + The expression $\hat{z}_{f;i} = \mathcal{E}^*(f)|_i$ represents the stochastic estimation of fixed point at $i^{th}$ iteration. For clarity and consistency we aim to change this to $\hat{z}_f^{(i)} = \mathcal{E}^*_f(\hat{z}_f^{(i-1)})$. This will represent the iterative nature of the stochastic estimator in a better way.
>
> We will update the paper with notations and better explanations upon acceptance.
>
> ---
>
> 3. In Table 2, we have shown the number of minimal states in Tomita grammars. The goal is to extract DFA with the number of states close to the optimal number of states required for a given language (minimalistic or minimal DFA). If the extraction process produces more states, that can happen for any of the following reasons:
> + RNN could not converge to the actual states.
> + The extraction process could not partition state space optimally.
>
> It is well known that converting any DFA (N) to minimal DFA (M`) can be achieved in polynomial time. However, it is important to note that NNs are stochastic thus, initially, state machines learned from RNN can be considered as non-deterministic or probabilistic. In this scenario, if we assume language learned by RNN (L1) = NFA (L1) and try to minimize NFA, then the problem becomes NP-complete or PSPACE-complete problem (Malcher et al., 2004). Furthermore, **the decidability problem for first-order RNN with probabilistic FA/ Weighted FA/non-deterministic FA is undecidable (marzouk et al., 2020)**. Furthermore, the equivalence between two DFAs is also NP-complete (Martens et al., 2023).
>
> Thus, in both cases mentioned above, even if we get a state machine or DFA, it would not be possible to extract the minimal and correct DFA in polynomial time. Thus it becomes very difficult to show empirically that two languages accepted by source DFA and RNN DFA are same. Given equivalence is NP-complete.
> To this end it’s important we have RNNs that could extract minimal DFA without any bell and whistle and we show second order RNN can stably achieve this.
>
> We motivate our paper and aim to showcase the empirical results in our Introduction. We provided R1 and R2 in a more theoretical perspective only after introducing the underlying mathematical basis and formulations in the Background and Methodology section.
>
> ---
>
> 4. The significance of this paper is in the context of the stability of rules extraction and embedding or RNN verification and explainability. We compare extraction methods for 1st and 2nd order RNNs. This paper encourages the research community to opt for RNN architectures that can estimate states more stably and use stable state extraction mechanisms for rule verification. This is an essential step in stable and explainable AI. We have covered the importance of this work in the first point.

---

> ### Author Response · Authors · 2023-11-28
>
> (Part 3/3)
>
> **References**
>
> Malcher, A., 2004. Minimizing finite automata is computationally hard. Theoretical Computer Science, 327(3), pp.375-390.
>
> Martens, J., 2023. Deciding minimal distinguishing DFAs is NP-complete. arXiv preprint arXiv:2306.03533.
>
> Marzouk, R. and de la Higuera, C., 2020. Distance and equivalence between finite state machines and recurrent neural networks: Computational results. arXiv preprint arXiv:2004.00478.
>
> Wang, Q., Zhang, K., Liu, X. and Giles, C.L., 2018. Verification of recurrent neural networks through rule extraction. arXiv preprint arXiv:1811.06029.
>
> Giles, C.L. and Omlin, C.W., 1993. Extraction, insertion and refinement of symbolic rules in dynamically driven recurrent neural networks. Connection Science, 5(3-4), pp.307-337.
>
> Delétang, G., Ruoss, A., Grau-Moya, J., Genewein, T., Wenliang, L.K., Catt, E., Cundy, C., Hutter, M., Legg, S., Veness, J. and Ortega, P.A., 2022. Neural networks and the chomsky hierarchy. arXiv preprint arXiv:2207.02098.
>
> Orvieto, A., Smith, S.L., Gu, A., Fernando, A., Gulcehre, C., Pascanu, R. and De, S., 2023. Resurrecting recurrent neural networks for long sequences. arXiv preprint arXiv:2303.06349.
>
> Bhattamishra, S., Ahuja, K. and Goyal, N., 2020. On the ability and limitations of transformers to recognize formal languages. arXiv preprint arXiv:2009.11264.
>
> Merrill, W., 2019. Sequential neural networks as automata. arXiv preprint arXiv:1906.01615.
>
> Joulin, A. and Mikolov, T., 2015. Inferring algorithmic patterns with stack-augmented recurrent nets. Advances in neural information processing systems, 28.
>
> Stogin, J., Mali, A. and Giles, C.L., 2020. A provably stable neural network Turing Machine. arXiv preprint arXiv:2006.03651.
>
> Das, S., Giles, C. and Sun, G.Z., 1992. Using prior knowledge in a NNPDA to learn context-free languages. Advances in neural information processing systems, 5.
>
> Alur, R. and Madhusudan, P., 2009. Adding nesting structure to words. Journal of the ACM (JACM), 56(3), pp.1-43
>
> Omlin, C. W., & Giles, C. L. (1996). Constructing deterministic finite-state automata in recurrent neural networks. Journal of the ACM (JACM), 43(6), 937-972.

---

> ### Author Response · Authors · 2023-11-29
> **A Gentle Reminder**
>
> Dear Reviewer 17Qn,
>
> Thank you for your positive review and valuable insights on our work. We have addressed all your comments/concerns in our detailed response and believe we have resolved all problems. Should any issues remain, we are ready to provide additional clarifications.
> As the rebuttal phase is nearing its deadline, we look forward to engaging in a timely discussion.
> Thank you again for your time and effort!
>
> Best regards,
> Authors

---

### Author Response · Authors · 2023-11-19

We thank and gratefully appreciate the reviews provided by the esteemed reviewers. Here, we provide an overview of our paper for better clarity.

### Paper Overview


(Giles, 1993) introduced the idea of extracting DFA from first and second-order RNNs. Subsequent works in the area focussed on DFA extraction for RNNs trained on regular languages like Tomita and counter languages like Dyck. (Omlin, 1996) shows the stability of second-order RNNs by proving bounds on the discriminant function using Browers’s fixed point theorem. (Stogin and Mali, 2020) use the same line of argument to establish the stability of Tensor Networks.

This is the first work to empirically study and contrast the stability of DFAs extracted from quantization and equivalence query-based methods. We train 4 different RNN models: LSTM, GRU, MIRNN, and O2RNN on Tomita and Dyck languages across 10 seeds and show stability of DFA extraction for quantization and equivalence query methods. LSTM and GRU are 1st order RNNs. O2RNN is a second-order RNN  while MIRNN is a low-rank approximation of 2nd-order RNN. (Merrill, 2020) have shown that LSTMs can model restricted counters while GRU is can model strictly regular languages. Although Dyck languages are counter languages, bounded dyck languages can be modeled by DFA with an increased number of states.

For quantization, we use k-means clustering and self-organizing maps. For the equivalence query-based method, we use L*, which was proposed by (Angluin, 1987). We train our networks with varying hidden state sizes and observe DFA performance and stability trends.

We ask two research questions:

**R1**. Does there exist a class of stable RNNs, that, for any regular language over dataset or distribution generated over any probability distribution ($\mathbb{P}_d$) clusters the internal states in reasonable time to approximate minimal DFA with high probability ($\mathbb{P}_m$), such that difference between states of RNN ($\hat{Q}$) and DFA ($Q$) in Euclidean space is within the small bound.
$||Q - \hat{Q}||_2^2 < \epsilon$

**R2**. RNNs that are partially trained and do not achieve perfect accuracy (i.e. $100 \%$) are known as partially trained RNNs. We consider them as a stochastic state estimator for that language. How do different DFA extraction methods affect optimal state estimation from partially trained RNN such that the extracted state ($\hat{Q}_e$) is much closer to the optimal state ($\hat{Q}$) of stable RNN.
    $||\hat{Q} - \hat{Q}_e||_2^2 < \epsilon_e$, where $\epsilon_e \geq \epsilon$

From our experiments for R1 we observe that O2RNN outperforms other RNN architectures to estimate stable optimal states of the minimal DFA of a given language. Also, quantization methods outperform equivalence query methods in estimating. We observe a similar trend while extracting DFAs from partially trained RNNs on Tomita grammars for R2. R2 is especially important for L* since it assumes Oracle has complete knowledge of the underlying DFA. This assumption breaks when we use partially trained RNNs as Oracle for DFA extraction. Quantization-based methods do not have this constraint and can even overcome small errors in state estimation by RNNs to produce stable DFAs.


### Why is this important?

DFAs are universally accepted as rule discriminators and acceptors and find implementation in a wide range of applications. It has been shown in previous works that DFAs can be used and are an essential tool for verification, explanation, and rule embedding in Recurrent Neural Networks. The stability of DFA extraction from RNNs becomes a necessary aspect and requires careful investigation. In this work, we provide empirical results for the stability of DFA extraction from different extraction algorithms on various RNN architectures.


### References:

DANA ANGLUIN. Learning regular sets from queries and counterexamples. INFORMATION AND COMPUTATION, 75:87–106, 1987

C Lee Giles and Christian W Omlin. Extraction, insertion and refinement of symbolic rules in dynamically driven recurrent neural networks. Connection Science, 5(3-4):307–337, 1993.

Christian W. Omlin and C. Lee Giles. 1996. Constructing deterministic finite-state automata in recurrent neural networks. J. ACM 43, 6 (Nov. 1996), 937–972.

John Stogin, Ankur Mali, and C Lee Giles. A provably stable neural network turing machine. arXiv preprint arXiv:2006.03651, 2020.

William Merrill, Gail Weiss, Yoav Goldberg, Roy Schwartz, Noah A. Smith, and Eran Yahav. A formal hierarchy of RNN architectures. In Proceedings of the 58th Annual Meeting of the Association for Computational Linguistics, pp. 443–459, Online, July 2020. Association for Computational Linguistics

---

### Author Response · Authors · 2023-11-28

Dear Reviewers nZtd, AfSv, and 17Qn,

We thank you for your very valuable insights and suggestions. We have now revised our manuscript to be more accessible to readers without a background in the stability theory of RNN, rule extraction, and the theory of computation. These topics are part of most undergraduate computer science and ML curricula. However, we agree with the reviewers that the topics have lately received less attention in the neural networks community. Thus, educating the audience to understand our contribution better is essential. Since the ICLR format has a strict page limit, we must balance the background and contributions, making it necessary to defer some details to the appendix. We believe our revised version has struck a balance and is accessible to the ICLR audience. We would like to hear from reviewers (provide names) whether our responses have resolved their queries and whether they think there are any open accessibility issues. We will be happy to engage in any discussion to clarify any concerns further.

Thank You!

---

### Meta-Review · Area_Chair_Yuag · 2023-12-21

**Metareview:**

After careful consideration of the reviewers' feedback and authors' responses, the consensus is to reject the paper from the conference. While the paper's attempt to empirically investigate the stability of rule extraction from trained RNNs is commendable, there are significant concerns that have influenced this decision.

Reviewer nZtd, while commending the paper's presentation, has expressed a lack of expertise in the area, which has limited their ability to thoroughly assess the paper's contributions and novelty. The reviewer has suggested that the paper would benefit from a clearer definition of stability and a stronger justification for the conclusions drawn. Although the authors have responded to these comments with a detailed explanation, the response seems to be more of a clarification of existing work rather than addressing the novelty or the broader impact of the study.

Reviewer AfSv has indicated that the paper could be a solid contribution; however, they pointed out the need for a more formal mathematical formulation. The authors have attempted to address this by adding a Stability Analysis section and updating the mathematical formulations. However, the response and modifications made during the rebuttal phase might not be sufficient to overcome the concerns raised by other reviewers.

Reviewer 17Qn has raised significant issues regarding the paper's narrow focus, the lack of clarity in the theoretical background, and the presentation of the key evaluation metrics. The reviewer also questions the overall significance of the results and their implications beyond RNNs and rule extraction approaches. Although the authors have provided a detailed rebuttal, it seems that the concerns regarding the narrow focus and the limited implications of the study remain unresolved.

The lack of engagement from the reviewers during the rebuttal session is also telling. It suggests that the revisions and responses provided by the authors did not sufficiently address the concerns raised, leaving the reviewers unconvinced of the paper's broader relevance and impact.

In light of these considerations, the paper does not meet the acceptance criteria for the conference. The decision to reject is based on the limited scope of the study, concerns about the novelty and broader impact, and the need for a clearer theoretical foundation and justification of the methods and results.

**Justification For Why Not Higher Score:**

The score for rejection is aligned with the concerns raised by the reviewers. A higher score would not be appropriate given the unresolved issues with the paper's focus, novelty, and contribution to the field. Moreover, the lack of engagement from the reviewers during the rebuttal session indicates that the paper did not inspire sufficient confidence in its significance.

**Justification For Why Not Lower Score:**

N/A

---

### Decision · Program_Chairs · 2024-01-16

Reject